# Western Diet Induced Remodelling of the Tongue Proteome

**DOI:** 10.3390/proteomes9020022

**Published:** 2021-05-12

**Authors:** Mriga Dutt, Yaan-Kit Ng, Jeffrey Molendijk, Hamzeh Karimkhanloo, Luoping Liao, Ronnie Blazev, Magdalene K. Montgomery, Matthew J. Watt, Benjamin L. Parker

**Affiliations:** Department of Anatomy and Physiology, School of Biomedical Science, The University of Melbourne, Melbourne, VIC 3010, Australia; mriga.dutt@unimelb.edu.au (M.D.); yaankitn@student.unimelb.edu.au (Y.-K.N.); jeff.molendijk@unimelb.edu.au (J.M.); hamzeh.karimkhanloo@unimelb.edu.au (H.K.); luopingl@student.unimelb.edu.au (L.L.); ronnie.blazev@unimelb.edu.au (R.B.); magdalene.montgomery@unimelb.edu.au (M.K.M.); matt.watt@unimelb.edu.au (M.J.W.)

**Keywords:** tongue, proteomics, obesity, taste, metabolism, high-fat diet

## Abstract

The tongue is a heavily innervated and vascularized striated muscle that plays an important role in vocalization, swallowing and digestion. The surface of the tongue is lined with papillae which contain gustatory cells expressing various taste receptors. There is growing evidence to suggest that our perceptions of taste and food preference are remodelled following chronic consumption of Western diets rich in carbohydrate and fats. Our sensitivity to taste and also to metabolising Western diets may be a key factor in the rising prevalence of obesity; however, a systems-wide analysis of the tongue is lacking. Here, we defined the proteomic landscape of the mouse tongue and quantified changes following chronic consumption of a chow or Western diet enriched in lipid, fructose and cholesterol for 7 months. We observed a dramatic remodelling of the tongue proteome including proteins that regulate fatty acid and mitochondrial metabolism. Furthermore, the expressions of several receptors, metabolic enzymes and hormones were differentially regulated, and are likely to provide novel therapeutic targets to alter taste perception and food preference to combat obesity.

## 1. Introduction

The mammalian tongue is a highly innervated, specialised muscular organ that is vital for speech, vocalisation and hosts the gustatory system, allowing for detection of flavour and taste. Taste perception occurs in taste buds, which are specialised sensory cells embedded in the stratified epithelium of the tongue, and classified as either Type I, II or III cells [1,2,3,4,5]. The glial-like type I cells maintain environmental homeostasis through elimination of extracellular neurotransmitters and ions [6]. Type II cells house the sensory G-protein coupled receptors (GPCR) used for chemodetection of sweet, bitter and the amino-acid based umami stimuli [7], while the Type III cells undergo activation of ion/ligand-gated ion channels to allow for sour and salty taste perception [8,9]. Altogether, these gustatory receptors provide a positive stimulus for appetite satiation and thus promote ingestion of foods [5]. Likewise, alterations in these chemodetection pathways may change an individual’s nutritional preference, thereby skewing the uptake of a balanced diet. Therefore, the tongue is highly relevant in understanding dietary diseases, predominantly obesity, a prevalent metabolic disease that typically involves excessive consumption of high calorie ‘Western diets’ rich in carbohydrates and fats.

Though taste is primarily categorised into sweet, salty, bitter, sour and umami, recent studies show mounting evidence that fat-induced activation of the gustatory system is being recognised as a ‘fatty’ taste [10]. While triglycerides have not been shown to stimulate taste cells, oral lipase activity in rodents was found to digest fats rapidly and to liberate free fatty acids (FFAs) in the immediate environment of the taste bud [11]. Although current knowledge on how fatty acids can activate the gustatory system is limited, several signalling candidates have been hypothesised. Some studies propose delayed rectifying K^+^ channels (DRKs) as a means for sensing fatty taste, as they are inhibited by polyunsaturated fatty acids (PUFA) in taste bud cells [12]. CD36 fatty acid translocase, a transmembrane glycoprotein, may also promote fat taste transduction, as it is highly expressed in circumvallate taste buds, and mice lacking CD36 displayed a reduced fat preference and intake [13,14]. Interestingly, CD36, DRKs and FFA receptors were found to be highly expressed in human fungiform papillae, further supporting their involvement in sensing the “fatty” taste [15]. Another hypothesis is a GPCR-mediated fatty taste signalling, specifically by GPR40 and GPR120 which are known to respond to FFAs with knock-out mouse lines displaying diminished preference for fatty acids [16]. While each of these candidate receptors may act as independent pathways of fatty taste transduction, it has been suggested that CD36 may instead serve as an acceptor molecule, binding FFAs and transporting them to GPCRs [17]. While downstream signalling remains largely undetermined, Transient receptor potential cation channel (TRPM5) and Toll-like receptor 4 (TLR4) are thought to be involved, as evidenced by deficits in fat taste when knocked out in mice [18,19].

There is growing evidence to suggest gustatory pathways and the enzymes processing food in the tongue are dynamically altered during consumption of calorie-rich diets or the progression of metabolic syndrome. Studies in rodents fed a high-fat diet (HFD) found altered levels of α-gustducin, phospholipase-C-β2 (PLCβ2) and CD36 without taste cell number being affected, suggesting an effect on taste transduction and tongue metabolism [20,21,22,23]. Diet intervention studies in humans revealed increased expression of free fatty acid receptor 4 (FFAR4) in fungiform papillae of subjects fed a low-fat diet [24]. The mechanism of these changes remains to be elucidated, but altered levels of hormones associated with metabolic syndrome, such as insulin, may have a direct effect on the tongue proteome. Indeed, exposure of 3D taste bud organoids to insulin decreases taste cell numbers and mRNA expression levels of certain sweet sensing genes such as taste receptor type 1 member 3 (T1R3), gustducin and carbonic anhydrase 4 (CA4) [25].

Despite this complex connection between the regulation of taste receptors and metabolism in the tongue, there has been no global quantitative analysis to characterise the mammalian tongue proteome and monitor changes following the consumption of Western diets. Here, we performed the most detailed proteomic analysis of mouse tongue to date and investigated dynamic remodelling following chronic exposure to a diet high in carbohydrate, fat and cholesterol. We defined the tongue-enriched proteome and identified hundreds of differentially regulated proteins which may uncover potential therapeutic targets within the tongue to combat obesity and other dietary diseases.

## 2. Materials and Methods

### 2.1. Animal Husbandry

All experimental animals were housed and maintained in compliance with the Australian code of practice for the care and use of animals for scientific purposes as set out by the National Health and Medical Research Council (Australia); all experimental procedures were conducted as per animal ethics approval number 2015115. Briefly, male C57BL/6J mice were housed within a humidity-controlled environment with temperatures maintained between 21 and 23 °C and subjected to a 12:12 h light:dark cycle. Mice were housed in filter-top cages with ad libitum access to either a standard rodent chow (5% energy from fat), or a Western diet comprising 40% energy from fat (including 18% trans-fats), 40% energy from carbohydrates (including 20% fructose), 18% protein and 2% cholesterol (SF16-033 Specialty Feeds, Glen Forrest, Australia). Following 32 weeks on chow (n = 7) or Western diet (WD; n = 7), mice were sacrificed and the oral section (demarcated from the apex to the root) of the tongue was rapidly dissected, frozen and stored at −80 °C.

### 2.2. Metabolic Assessment and Data Analysis

Mice were fasted for 4 h, then gavaged orally with liquid glucose (2 g/kg body weight). Blood glucose levels were measured at 0, 15, 30, 45, 60, 90 and 120 min using the Accu-Chek II glucometer (Roche Diagnostics, Basel, Switzerland). Time domain nuclear magnetic resonance (TD-NMR) was performed for fat and lean mass measurements using the Bruker LF50H Minispec Body Composition Analyser (Bruker, Billerica, MA, USA) in accordance with the manufacturer’s instructions. Comparison of chow vs. Western diet data for lean and fat mass measurements was performed with 2-way ANOVA with Bonferroni post hoc analysis and unpaired Student’s 2-way *t*-tests, respectively. Statistical analyses and data visualisation for all experiments were performed using Prism v9.0 (GraphPad, San Diego, CA, USA), and data were plotted as the standard error of the mean (SEM).

### 2.3. Proteomics Sample Preparation

All solvents and chemicals were of LC-MS grade and were purchased from Merck (Kenilworth, NJ, USA), unless stated otherwise. Frozen tongue tissues were denatured in 600 µL guanidine-HCL lysis buffer (6 M guanidine hydrochloride, 100 mM Tris pH 8.5, 10 mM tris (2-carboxyethyl)phosphine, 40 mM 2-chloroacetamide) via tip-probe sonication (2 × 30 s cycles, power setting 20%, QSonica, Newtown, CT, USA), heated at 95 °C for 5 min, subsequently centrifuged at 18,000× *g* for 10 min at 4 °C to remove cellular debris and the supernatant was collected. Then, 500 µL of the supernatant was diluted 1:1 with water, and 4 volumes of chilled acetone was added to each sample to precipitate protein overnight at −20 °C. The samples were centrifuged at 4000× *g* for 5 min at 4 °C to collect the protein pellet. The protein pellet was washed with 4 mL of cold 80% *v*/*v* acetone in water, subjected to brief tip-probe sonication, and centrifuged at 4000× *g* for 5 min at 4 °C. The recovered protein pellet was air-dried to remove residual acetone and subsequently resuspended in 500 µL of digestion buffer (10% *v*/*v* 2,2,2 trifluoroethanol in 100 mM HEPES, pH 7.4). Sample protein concentrations were determined using the Pierce BCA protein assay kit (Thermo Fischer, Waltham, MA, USA), normalised to 40 µg/20 µL and digested with 0.8 µg each of sequencing grade trypsin (Sigma, St. Louis, MO, USA) and LysC (Wako, Osaka, Japan) overnight at 37 °C with 1500 rpm shaker conditions. The digested peptides were acidified with 1% *v*/*v* trifluoroacetic acid (TFA) and desalted using styrene divinylbenzene-reversed-phase sulfonate (SDB-RPS, packed in-house) stage tips. The peptides were eluted with 80% *v*/*v* acetonitrile in 5% *v*/*v* ammonium hydroxide, dried under vacuum, resuspended in 2% *v*/*v* acetonitrile in 0.1% *v*/*v* TFA and stored at −80 °C. Approximately 20 µg of pooled peptides, prepared by combining n = 3 digested samples for each diet, were fractionated with a Dionex 3500 on a 30 cm × 320 µm column packed in-house with C18BEH (3 µm; Waters, Milford, MA, USA) over 72 min at a flow rate of 6 µL/min. The peptides were eluted over a linear gradient of 3–40% Buffer B (Buffer A: 10 mM ammonium formate pH 7.9; Buffer B: 90% *v*/*v* acetonitrile) with a total of 72 fractions collected which were automatically concatenated in a looping fashion into 12 fractions, dried by vacuum centrifugation and resuspended in 2% *v*/*v* acetonitrile in 0.1% *v*/*v* TFA and stored at −80 °C.

### 2.4. Liquid Chromatography and Tandem Mass Spectrometry

Peptides were analysed on a Dionex 3500 nanoHPLC, coupled to an Orbitrap Eclipse mass spectrometer (Thermo Fischer, Waltham, MA, USA) via electrospray ionization in positive mode with 1.9 kV at 275 °C. Separation was achieved on a 40 cm × 75 µm column packed with C18AQ (1.9 µm; Dr Maisch, Ammerbuch, Germany) (PepSep, Marslev, Denmark) over 120 min at a flow rate of 300 nL/min. The peptides were eluted over a linear gradient of 3–40% Buffer B (Buffer A: 0.1% formic acid; Buffer B: 80% *v*/*v* acetonitrile, 0.1% *v*/*v* FA) and the column was maintained at 50 °C. For data-dependent acquisition (DDA) on pooled, fractionated samples, an MS1 spectrum was acquired over the mass range 350–1550 *m*/*z* in the orbitrap (120,000 resolution, 1 × 10^6^ automatic gain control (AGC) and 50 ms maximum injection time) followed by MS/MS analysis with fixed cycle time of 3 s via HCD fragmentation mode and detection in the orbitrap (15,000 resolution, 1 × 10^5^ AGC, 22 ms maximum injection time, 1.2 *m*/*z* isolation width, 10 ppm dynamic exclusion for 60 sec). For data-independent acquisition (DIA) on individual samples (n = 7 for each diet), an MS1 spectrum was acquired over the mass range 350–1200 *m*/*z* in the orbitrap (120,000 resolution, 1 × 10^6^ AGC and 50 ms maximum injection time) followed by MS/MS analysis via HCD and detection in the orbitrap (30,000 resolution, 1 × 10^6^ AGC, 54 ms maximum injection, 36 × 24 *m*/*z* isolation windows with 1 *m*/*z* overlap).

### 2.5. Proteomic Data Analysis

DDA data were searched against the UniProt mouse database (June 2020; UP000000589_109090 and UP000000589_109090_additional; 63,438 entries). DDA data were analysed with MaxQuant v1.6.12.0 using default parameters with peptide spectral matches and protein false discovery rate (FDR) set to 1% [26]. Data were searched with oxidation of methionine set as the variable modification and carbamidomethylation set as the fixed modification. First search MS1 tolerance was set to 20 ppm followed by recalibration and main search MS1 tolerance set to 4.5 ppm, while MS/MS tolerance was set to 20 ppm. DIA data were analysed with Spectronaut v14.7.201007.47784 using default parameters with precursor and protein FDR set to 1% (Biognosys, Schlieren, Switzerland). Quantification was performed at the MS2 level with 3–6 fragment ions based on AUC. Gene ontology annotation, multivariate and statistical analyses were performed in Perseus v1.6.10.50 [27]. Comparison of chow vs. Western diet proteomes was performed with unpaired Student’s 2-way *t*-tests with permutation-based FDR correction and significant proteins assigned based on q < 0.05, and data visualisation was done using Prism v9.0. (GraphPad, San Diego, CA, USA). The tongue enriched proteome was compared to The Human Protein Atlas [28] and network analysis was performed in STRING v11.0 [29].

## 3. Results

### 3.1. The Mouse Tongue Proteome

To characterise the oral section of the mouse tongue proteome, we performed 2D-nano liquid chromatography coupled to tandem mass spectrometry (2D-nLC-MS/MS) analysis of LysC/tryptic peptides by data-dependent acquisition. A total of 5904 unique protein groups were identified (4604 unique protein groups with at least two unique peptides) (Appendix A Appendix A). Gene Ontology cellular compartment analysis annotated 73% of the proteome to the cytoplasm, with more than 1000 proteins annotated as plasma membrane-associated, and 974 and 624 annotated to the nucleus and endoplasmic reticulum/sarcoplasmic reticulum (ER/SR), respectively (Figure 1a). A further 568 proteins were annotated as mitochondrial, with several other subcellular annotations represented, including contractile fibre, extracellular matrix and the lysosome. To our knowledge, this is the most comprehensive proteomic characterisation of the mammalian tongue spanning a range of Gene Ontology molecular functions with hundreds of proteins annotated as receptors, DNA/RNA-binding, transporters or containing ATPase/oxidoreductase activity (Figure 1b). We next compared our proteomic data normalized via intensity-based absolute quantification (iBAQ) [30] to mRNA transcriptome data of human tongue available through the Human Protein Atlas: Tissue Atlas (http://www.proteinatlas.org; accessed on 1 March 2021) [28]. We observed an overall significant correlation (r^2^ = 0.22, *p* = 6.5 × 10^−23^) consistent with previous studies despite species differences (Figure 1c). Finally, we investigated the expression of proteins enhanced in the tongue compared to other tissues, as these are more likely to play important roles in tongue biology. Data were normalized to protein/nucleic acid deglycase DJ-1 (PARK7), a ubiquitously and highly expressed protein previously found to have the lowest variability in abundance among different cell/tissue types in human and mouse [31]. Normalised data were compared to a previous mouse proteomic tissue atlas in which PARK7 was quantified in every sample [32] and enhanced proteins defined as four-fold higher in abundance compared to the average in all other tissues as previously described in The Human Protein Atlas. This analysis identified 57 enhanced proteins, with alpha-actin (ACTA1), titin (TTN) and creatine kinase M-type (CKM) as the top three most abundant (Figure 1d). As expected, the most similar tissue was hind-limb skeletal muscle, but interestingly, we identified three proteins elevated specifically in tongue, including four and a half LIM domains protein 1 (FHL1), junction plakoglobin (JUP) and periplakin (PPL). Our data will be a rich resource to investigate the proteomic landscape of the mouse tongue.

### 3.2. Quantification of the Tongue Proteome in Response to Western Diet

To investigate how the tongue proteome is regulated following chronic consumption of a calorie-rich diet, C57BL/6J mice were fed a normal chow or Western diet high in fat, sugar and cholesterol for 32 weeks (n = 7/diet). Western diet induced an almost 50% increase in body weight (Figure 2a), which was associated with a significant increase in percentage fat mass and decrease in lean mass, as assessed by TD-NMR (Figure 2b). As expected, mice fed the Western diet had higher fasting blood glucose levels and were significantly glucose intolerant as assessed by an oral glucose tolerance test (Figure 2c). Quantification of the tongue proteome was performed by analysing LysC/tryptic peptides by 1D-nLC-MS/MS coupled to data-independent acquisition (Figure 3a) (Appendix A). A total of 5295 protein groups were quantified with 5238 identified in all 14 samples (4329 unique protein groups with at least two peptides) (Figure 3b). Principal component analysis and unsupervised hierarchical clustering revealed robust quantification and segregation of animals based on their dietary interventions (Figure 3c,d). A total of 220 proteins were significantly regulated between the diets with 146 up- and 74 down-regulated proteins, respectively (Figure 3e) (q < 0.05, *t*-test with Benjamini–Hochberg multi-hypothesis testing adjustment). The most significantly up-regulated protein was the antioxidant peroxidasin (PXDN), while alpha-1-antitrypsin (SERPINA1E) was the most significantly decreased protein. To gain an overview of the regulated cellular pathways, enrichment analysis was performed against the Kyoto Encyclopedia of Genes and Genomes (KEGG) database [33]. The most up-regulated pathways increased with Western diet were those associated with fatty acid and branch chain amino acid metabolism, including proteins associated with the peroxisome, while proteins involved in complement and coagulation cascades were the most down-regulated (Figure 3f) (q < 0.05, Fisher’s exact test with Benjamini–Hochberg multi-hypothesis testing adjustment). Figure 3g focuses on the enzymes of the fatty acid degradation pathway, which were all up-regulated following chronic consumption of a calorie-rich Western diet.

### 3.3. Comparison of Proteome Changes of the Mouse Tongue and Skeletal Muscle Following Consumption of a Western Diet

We next compared changes in the tongue proteome in response to a Western diet with a previously published proteomic analysis of gastrocnemius skeletal muscle in response to a similar but shorter high-fat diet intervention [34]. A total of 29 proteins were significantly regulated (28 up- and 1 down-regulated) in both muscles and all were regulated in the same direction (Figure 4a). Examples include the up-regulation of pyruvate dehydrogenase kinase-4 (PDK4), which phosphorylates and inhibits pyruvate dehydrogenase complex, thereby reducing carbohydrate oxidation and increasing fatty acid metabolism, delta(3,5)-delta(2,4)-dienoyl-CoA isomerase (ECH1), which is involved in fatty acid β-oxidation, and mitochondrial uncoupling protein 3 (UCP3), which regulates anion transport and energy production. Interestingly, the only protein found to be downregulated was malectin (MLEC), which regulates quality control of newly synthesised glycoproteins in the ER/SR, but its functional role in response to dietary adaptations remains unknown. Interestingly, more than 180 and 80 proteins were significantly regulated in either skeletal muscle or tongue, respectively. To investigate these differences, we next performed protein–protein association analysis, which included the annotation of either direct or indirect protein–protein interactions or previously characterised co-expression. This analysis revealed a tight network of proteins associated with fatty acid oxidation regulated in both muscles (Figure 4b). The skeletal muscle network regulated with HFD was dominated by a tight cluster of mitochondrial complex I proteins, tricarboxylic acid cycle (TCA) proteins, several enzymes involved in lipid metabolism, protein synthesis and several other networks such as proteins involved in redox detoxification (Figure 4c). In contrast, the tongue network regulated with the Western diet included proteins involved in branch-chain amino acid metabolism, several apolipoproteins and chaperones, and enzymes involved in ubiquinone biosynthesis (Figure 4d). Thus, our integrative analysis provides an exciting glimpse into adaptations in striated muscle in response to a calorie-rich diet.

### 3.4. Regulation of Gustatory Systems in the Tongue Proteome in Response to Western Diet

We next investigated whether a Western diet, with elevated levels of fat, sugar and cholesterol, may change components of the sensory systems, specifically the gustatory system. Here, we annotated sensory systems using various databases, including UniProt Keywords, Reactome, KEGG and Gene Ontology. This analysis quantified >100 proteins which included not only proteins annotated as being involved in taste pathways, but also various sensory systems such as mechano-transduction, neuronal/pain sensing, olfactory, and sound sensing proteins, as many of these pathways share important downstream regulators. Notably, bitter sensing zinc-alpha-2-glycoprotein (AZGP1; Figure 5a), and fatty taste sensing fatty acid translocase (CD36; Figure 5b), were significantly upregulated in the tongue of the Western diet mice, indicating an enhanced perception/preference for these tastes in mice exposed long term to high levels of dietary fat, sugars and cholesterol. The inositol 1,4,5-triphosphate receptor 3 (ITPR3), a calcium transporter and critical receptor for sweet, bitter and umami tastes, was quantified, and its levels were unaffected by diet intervention (Figure 5c). However, several other proteins involved in inositol triphosphate (IP3) and calcium signalling were significantly decreased in response to the Western diet, including the sodium/hydrogen exchanger 1 (SLC9A1; Figure 5d), the ERM exchange cofactor that links SLC9A activation (SLC9A3R1; Figure 5e) and the cytosolic phosphoplipase A2 beta (PLA2G4B; Figure 5f). We also identified 40 secreted proteins differentially regulated in the tongue following chronic consumption of the Western diet (Figure 5g). This included the upregulation of several proteins involved in oxidation/reduction reactions, suggesting that the tongue may be under oxidative stress in response to the Western diet. We also observed the regulation of 16 proteins with peptidase activity or involved in the complement cascade, suggesting changes in protease processing of the tongue proteome. Taken together, our data provide new insights into the changes in the gustatory system in the mouse tongue in response to chronic consumption of a Western diet.

## 4. Discussion

Obesity is a complex disease that is driven by a combination of several genetic and lifestyle factors including excess consumption of calorie-rich diets and sedentary behaviour [35]. There is growing evidence that diet-induced obesity can alter the oral gustatory pathways, thus affecting nutritional intake and further exacerbating the condition [22,36,37,38]. Though several system-wide studies have investigated the effects of HFD consumption on whole body metabolism, none have reported on any associated changes occurring in the orosensory pathways [39,40,41,42]. Therefore, in this study, we have performed the first in-depth characterisation of the mouse tongue proteome following chronic intake of a typical Western diet, enriched in dietary fats, complex sugars and cholesterol. Using multi-dimensional separation techniques coupled to tandem mass spectrometry, we have revealed the landscape of the mouse tongue proteome and identified several key regulatory pathways that are remodelled following long-term exposure to Western diet.

Despite the functional relevance of the tongue in craniofacial and oral systems, there exists a knowledge gap in the representation of the tongue proteins in the Human Protein Atlas [28]. The current database includes transcriptomic, genomic and antibody-based characterisation of the tongue proteins [43,44] that is limited and does not represent the dynamically altered tongue proteins. Through this study, we have provided the first comprehensive mass spectrometry-based annotation of the mouse tongue proteome enriched with >5900 proteins, including a subset of proteins altered as a response to dietary stimuli. This dataset included hundreds of proteins involved in receptor- and transporter-mediated pathways and calcium binding, confirming the function of the tongue as a gustatory organ, mediating taste transduction through specialised taste receptors housed within Type II and III taste cells [7,45,46,47]. Likewise, a significant percentage of the proteome was implicated in nucleotide binding and oxidoreductase activities, implying a likely function played by the tongue proteins in energy metabolism. Furthermore, comparison of our data with the murine tissue protein map [32] identified elevated expression levels of a cluster of tongue proteins, specifically the cytoskeletal protein FHL1, and the adhesion proteins JUP and PPL (Figure 1d). FHL1 acts as a scaffolding protein during muscle assembly and promotes myoblast differentiation [48], while being implicated in the pathogenesis of several myogenic conditions, including muscular dystrophy and the tropical disease chikungunya [49,50]. JUP is expressed within intercellular junctions and desmosomes, critical for cell adhesion, and has been shown to promote metastasis in oral squamous cell carcinoma [51,52]. Desmosome-associated PPL was found to be expressed in neonatal mouse tongue [53]. As expected, myoproteins such as actin (ACTA1), titin (TTN), alpha-actinin 2 (ACTN2) and myosin 8 (MYH8) were also significantly enriched within the tongue proteome, as previously reported [28]. Therefore, our tongue proteomics data are a rich resource for investigating oral or muscle-related conditions and the complexity of the proteome further reiterates that the tongue is a highly intricate muscular tissue that warrants its own characterisation.

To further unravel the molecular link between a calorie-rich diet and the tongue tissue architecture, we administered a Western diet enriched with 40% fat energy, to mice over 32 weeks of age, which is considered equivalent to ~20 years in humans. A significantly longer timeframe was selected over previously published studies that ranged between 10 h and 11 weeks [21,22,23,36,54], thereby allowing us to model chronic human overconsumption. Interestingly, the Western diet appeared to have dramatically altered the tongue to exhibit a “fatty acid” centric proteomic profile (Figure 3f), as significant enrichment of proteins and enzymes involved in fatty acid processing was observed, including elongation, degradation, metabolism and peroxisome-mediated oxidation, akin to the changes in the skeletal muscle proteome under similar dietary conditions (Figure 4). This suggests that the tongue functions as an integral part of the musculoskeletal system and when challenged by increased dietary fat intake, it upregulates the feedback mechanism between lipid degradation and TCA pathways to mediate energy release in muscles, while attempting to reduce the excessive fatty acid reserves. Indeed, this is evidenced by the pronounced expression of the mitochondrial kinase PDK4 in the tongue exposed to a Western diet (Figure 4a). PDK4 is critical for metabolic substrate switching; that is, reducing carbohydrate oxidation and reciprocally increasing fatty acid oxidation to match energy requirements with the dominant metabolic substrate. Moreover, detection of these fatty acid processing proteins further confirms the role of the tongue in breaking down food particles allowing nutrient uptake and initiating digestion. Additionally, an upregulation of proteins regulating oxidation and bioavailability of metal ions was observed in the Western diet tongue proteome (Figure 3e). For example, peroxidasin homolog (PXDN) assists in extracellular matrix organisation and induces antimicrobial defence [55,56]; however, the association between PXDN and diet-induced obesity remains unknown. Similarly, an increase in heme protein hemopexin (HPX) and cuproprotein ceruloplasmin (CP) levels has been associated with increased triglyceride levels and obesity, respectively [57,58].

Finally, we evaluated whether the long-term chronic diet overconsumption affected the gustatory pathways of the tongue. The Western diet tongue proteome exhibited significant expression of the taste sensing proteins, fatty acid translocase (CD36) and zinc-alpha-2 glycoprotein (AZGP1) (Figure 5a,b). It was exciting to observe a positive correlation of CD36 with HFD as it is a critical fatty taste sensor which is localised within mammalian taste buds [13,15]. While speculative, prolonged intake of HFD may increase the sensitivity of the taste cells to fatty taste perception, which might in turn act as a positive taste stimulus in DIO to consume more dietary fats. This possibility warrants further investigation. While high expression of AZGP1 in Western diet mice agrees with its known functions in lipid degradation processes and impaired insulin sensitivity in obesity [59,60], AZGP1 was also suggested to be involved in bitter taste perception by Gene Ontology analyses. While the administered Western diet in this study was devoid of any bitterants, it is likely that the long-term consumption of a diet with saturated fat and sweet levels may reduce the sweet tasting cells with no change in the bitter sensing Type II taste receptors [36]. Therefore, an increase in AZGP1 might reflect an enhanced bitter perception due to the corresponding reduction in the sweet sensors and hence its putative role in gustation should be further investigated. Besides these, we did not detect other proteins directly implicated in sensing the classical five tastes; however, our data identified basal-level expression of several voltage-gated ion channels and transporters (Appendix A) that may be involved in taste sensing through intracellular Ca^2+^ mediated responses, thus indicative of the depth of our proteomics coverage. Thus, these data provide a new resource to further explore the putative changes in the gustatory system following high-fat diet consumption.

## 5. Conclusions

In summary, this study demonstrates the tongue to be an integral metabolic muscle with an emerging role in studying obesity. Diet-induced alterations in the lingual metabolic processes provide a unique perspective to further understand the underlying molecular mechanisms in the development of obesity and can also provide insights into the factors affecting the nutritional choices of obese individuals. Knowledge of these processes could eventually serve as a template for designing novel therapeutics that might positively influence nutritional choices of patients and help combat obesity.

## Figures and Tables

**Figure 1 proteomes-09-00022-f001:**
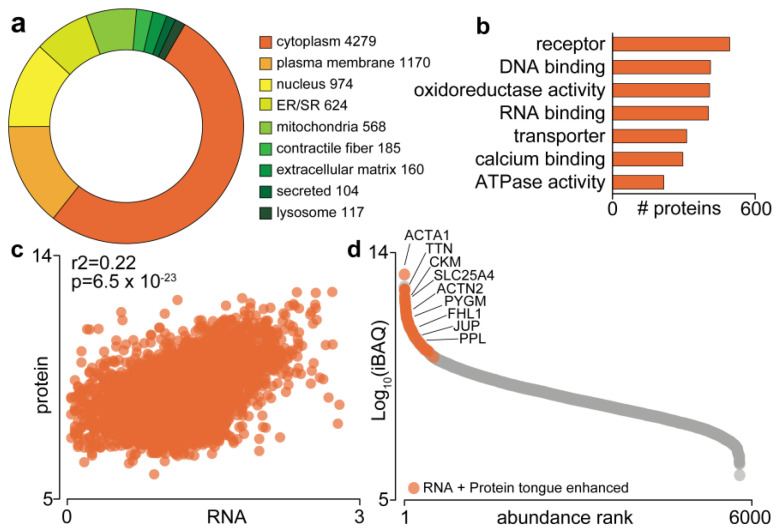
The mouse tongue proteome. (**a**) Gene Ontology subcellular distribution; (**b**) Gene Ontology molecular function; (**c**) correlation to transcriptomics data from the Human Protein Atlas; (**d**) Ranked abundance and enriched proteome. Alpha-actin (ACTA1); titin (TTN); creatine kinase M-type (CKM); ADP/ATP translocase 1 (SLC25A4); alpha-actinin-2 (ACTN2); glycogen phosphorylase, muscle form (PYGM); four and a half LIM domains protein 1 (FHL1); junction plakoglobin (JUP) and periplakin (PPL).

**Figure 2 proteomes-09-00022-f002:**
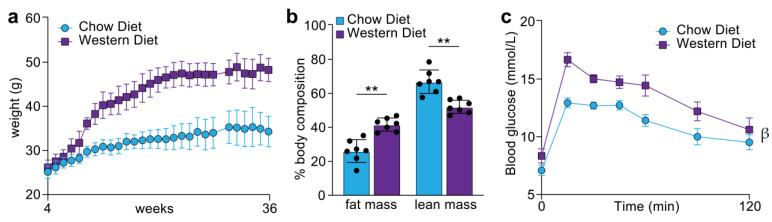
Whole-body metabolic assessments of chow and Western diet-fed mice. (**a**) Body weights; (**b**) body composition; (**c**) oral glucose tolerance test (oGTT). n = 7; ** *p* < 0.01 (students *t*-test); ^β^ *p* < 0.05 (two-way ANOVA); error bars SEM.

**Figure 3 proteomes-09-00022-f003:**
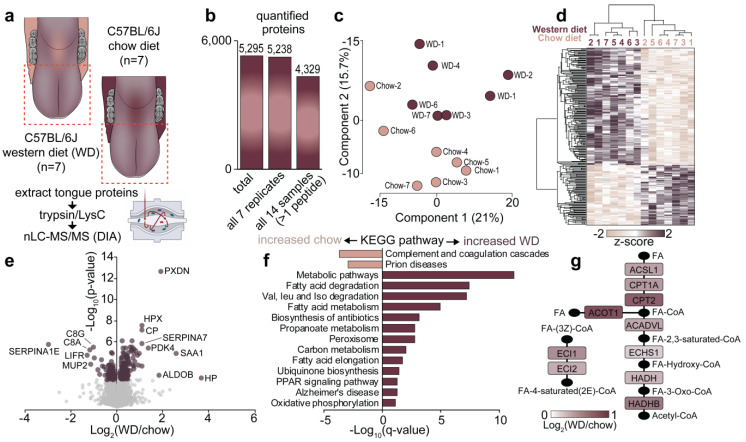
Quantification of the tongue proteome in response to Western diet. (**a**) Overview; (**b**) number of proteins quantified; (**c**) principal component analysis; (**d**) unsupervised hierarchical clustering; (**e**) volcano plot; (**f**) Kyoto Encyclopedia of Genes and Genomes (KEGG) pathway enrichment; (**g**) fatty acid degradation pathway.

**Figure 4 proteomes-09-00022-f004:**
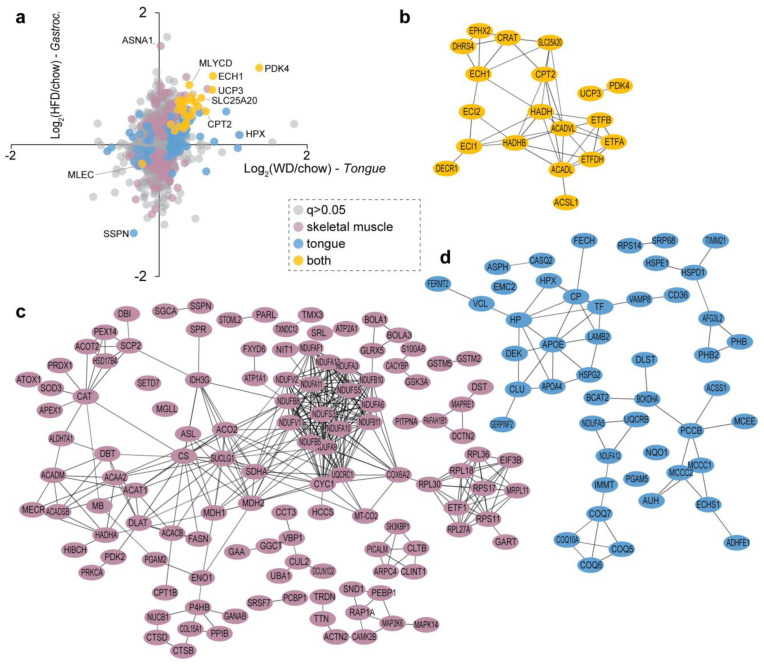
Quantification of mouse tongue and skeletal muscle in response to a Western diet and high-fat diet, respectively. (**a**) Scatter plot showing the distribution of proteins quantified in both tissues and regulated (q < 0.05). Network analysis of proteins regulated in (**b**) both tissues, (**c**) skeletal muscle only or (**d**) tongue only.

**Figure 5 proteomes-09-00022-f005:**
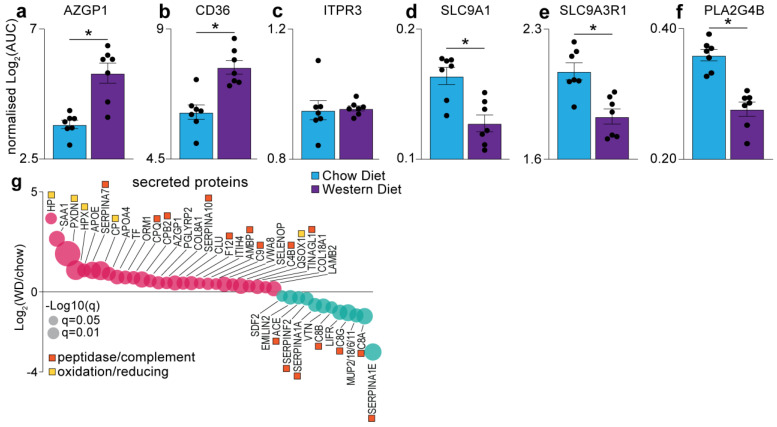
Quantification of gustatory system highlighting indicated proteins (**a**–**f**) and secreted proteins (**g**) following Western diet consumption. n = 7; * q < 0.05 (students *t*-test with permutation-based FDR correction); error bars SEM.

## Data Availability

The mass spectrometry proteomics data have been deposited to the ProteomeXchange Consortium via the PRIDE [61] partner repository with the dataset identifier PXD025190.

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
