# Peer review of "Western Diet Induced Remodelling of the Tongue Proteome"

_proteomes, 2021, doi:10.3390/proteomes9020022_

Round 1
Reviewer 1 Report
The authors show that the tongue is an integral metabolic muscle that plays a new role in the study of obesity. Diet-induced changes in lingual metabolic processes provide a unique perspective to further understand the underlying molecular mechanisms in the development of obesity and may also provide insight into the factors that influence dietary choices in obese individuals. Knowledge of these processes could ultimately serve as a template for the development of novel therapeutics that could positively influence patients' dietary choices and help combat obesity. So, this research paper can be published.
Perhaps, I would add a resume figure.
-Material and methods should be clarified: for analysis pooled and not individual?
-Data validation by Western blot should be attempted?
Reviewer 2 Report
The authors sought to characterize the proteomic landscape of the mouse tongue and quantified changes following chronic consumption of a chow or Western diet. I felt that a lot of effort was contributed to achieve this goal, and this manuscript is clear logic and well-written. However, it is mandatory to explain/correct the manuscript in some points.
- How many biological replicates were performed in the proteomic study?
- Has the p-value been corrected?
Reviewer 3 Report
The current manuscript focuses on the alterations in the tongue proteome induced by a high-fat, high-sugar Western diet in mice. This study is very relevant with global obesity reaching epidemic proportions due to consumption of energy-dense and processed foods, often referred to as the “Western diet”. The authors use mice on a prolonged diet of chow and western diet for their proteome analysis. The authors use a high number of animals for each group (n=7), which is a good number for a study like the present one, due to the high variance of individual response one can observe to an external factor like a diet. However, I think that a time-point comparison of proteome changes could have been a more interesting approach to study the progressive impact of a western diet on the global gustatory protein profile.
This study is descriptive and looks at a large number of proteins across the proteome. But an analysis of the tongue samples using tissue histology or western blots for some of the important proteins, which were differentially regulated between the two groups of animals validates the results further. The authors show gene ontology and PCA data in their results section. But the methods section does not mention any details about the software package they used to perform the ontology and the PCA analysis. Also, I do not find the two animal groups forming distinct clusters in the PCA on either of the Principal components. I would suggest that the authors look into other principal components which segregate their two animal groups, as the factors driving this segregation are the proteins that are contributing to the highest variance between the groups.
Typos:
Line 310: associated changes occouring -> occurring
The authors use proteomics to address a less explored aspect of tongue remodeling by the western diet. A comprehensive study like the current one is extremely useful for development of therapeutic targets for a wide range of lifestyle diseases affecting our modern world. I recommended the manuscript be accepted after a minor revision.
